# The Role of Gut Microbiome in Irritable Bowel Syndrome: Implications for Clinical Therapeutics

**DOI:** 10.3390/biom14121643

**Published:** 2024-12-21

**Authors:** Yucui Zhao, Shixiao Zhu, Yingling Dong, Tian Xie, Zhiqiang Chai, Xiumei Gao, Yongna Dai, Xiaoying Wang

**Affiliations:** 1Key Laboratory of Pharmacology of Traditional Chinese Medical Formulae, Ministry of Education, Tianjin University of Traditional Chinese Medicine, Tianjin 301617, China; zhaoyc12242022@163.com (Y.Z.); zsx1230224@163.com (S.Z.); tjutcmdyl@163.com (Y.D.); xt22640@163.com (T.X.); chzq20000@163.com (Z.C.); gaoxiumei@tjutcm.edu.cn (X.G.); 2School of Chinese Materia Medica, Tianjin University of Traditional Chinese Medicine, Tianjin 301617, China; 3State Key Laboratory of Component-Based Chinese Medicine, Tianjin University of Traditional Chinese Medicine, Tianjin 301617, China

**Keywords:** IBS, gut microbiome, diet-mediated, brain–gut axis, immune responses

## Abstract

Irritable bowel syndrome (IBS) is a functional gastrointestinal disorder (FGID) characterized by chronic or recurrent gastrointestinal symptoms without organic changes, and it is also a common disorder of gut–brain interaction (DGBIs).. The symptoms of IBS not only affect the quality of life for individual patients but also place a significant burden on global healthcare systems. The lack of established and universally applicable biomarkers for IBS, along with the substantial variability in symptoms and progression, presents challenges in developing effective clinical treatments. In recent years, preclinical and clinical studies have linked the pathogenesis of IBS to alterations in the composition and function of the intestinal microbiota. Within the complex microbial community of the gut, intricate metabolic and spatial interactions occur among its members and between microbes and their hosts. Amid the multifaceted pathophysiology of IBS, the role of intestinal microenvironment factors in symptom development has become more apparent. This review aims to delve into the changes in the composition and structure of the gut microbiome in individuals with IBS. It explores how diet-mediated alterations in intestinal microbes and their byproducts play a role in regulating the pathogenesis of IBS by influencing the “brain-gut” axis, intestinal barrier function, immune responses, and more. By doing so, this review seeks to lay a theoretical foundation for advancing the development of clinical therapeutics for IBS.

## 1. Introduction

IBS is known as a type of FGID [1]. It is also recognized as a prevalent disorder in the spectrum of DGBIs. The clinical presentation of IBS typically encompassed irregular bowel movements (changes in the frequency and consistency of stool), along with persistent and intermittent abdominal pain and bloating, all of which significantly impact the patient’s quality of life [2].

According to the Rome IV diagnostic criteria, IBS is categorized into four subtypes based on the predominant bowel habit pattern: IBS with constipation (IBS-C), IBS with diarrhea (IBS-D), mixed IBS with both constipation and diarrhea (IBS-M), and unclassified irritable bowel syndrome [3,4]. The symptoms associated with IBS not only diminish the quality of life for individual patients but also escalate the strain on the global healthcare system due to unnecessary or frequent medical visits and treatment costs [5].

Currently, the pathophysiological mechanisms underlying abdominal pain and changes in bowel habits in IBS remain incompletely understood, posing challenges to the development of effective treatment strategies for patients with this condition.

Within the intricate pathophysiology of IBS, the significance of intestinal microenvironmental factors in symptom development have become increasingly evident. This breakthrough has propelled the advancement of novel treatment approaches that specifically target the abnormal intestinal microenvironment. These strategies encompass antibiotics, prebiotics, probiotics, synbiotics, fecal microbiota transplantation (FMT), dietary modifications, and more [6]. The gastrointestinal tract (GI) represents a complex milieu populated by diverse microorganisms. Within this environment, intricate metabolic and spatial interplays occur among members of the intestinal microbial community as well as between microorganisms and the host. Studies have indicated that the microorganisms residing in the human intestine play a pivotal role in host metabolism. The intestinal microbiota has evolved into a real organ with distinct functions. It maintains a symbiotic relationship with the host and is viewed as a promising source of innovative therapies [7]. Gut microbes extract energy and nutrients from ingested foods while shielding hosts from pathogens. Several studies have underscored the significant impact of alterations in gut microbial composition and function on IBS pathophysiology [8]. This review primarily delves into the structural modifications of intestinal microbiota in IBS patients and the involvement of intestinal microbiota and its byproducts in modulating the pathogenesis of IBS through alterations in the “brain-gut” axis, intestinal barrier function, immune response, etc., thereby offering deeper insights into the disease’s pathogenesis.

IBS affects approximately 9.2% of the global population [9], can occur in all age groups, and is more common in women. The occurrence of IBS varies significantly across different countries and cultures [1]. The direct medical expenses associated with IBS are substantial. For instance, in the UK, annual direct medical costs for IBS can reach as high as CNY 1.14 billion to CNY 1.76 billion (based on an exchange rate of GBP 1 = CNY 8.74) [10]. In European countries, the total cost (direct and indirect) linked to IBS amounts to up to CNY 62.4 billion (based on an exchange rate of EUR 1 = CNY 7.8) [11]. Similarly, in China, the management of IBS incurs a total cost of approximately CNY 12.383 billion [12]. The economic impact is further compounded by the interference of IBS symptoms with daily activities and productivity, leading to notably higher annual expenses for patients experiencing severe gastrointestinal symptoms and psychological conditions [13]. Research has indicated that 82% of IBS patients experience an overall decline in work performance and daily functioning due to clinical symptoms, with 5–50% necessitating time off work because of their symptoms [14]. Moreover, the social and travel limitations imposed by IBS symptoms can impede understanding and support from friends, family, and healthcare providers, further diminishing the patient’s quality of life.

Delays in diagnosis and effective treatment initiation may exacerbate anxiety and depression, ultimately increasing mortality rates. Hence, there is an urgent need to assist IBS patients in accessing comprehensive care approaches to enhance their long-term quality of life and overall well-being.

## 2. Structural Characteristics of IBS Microbiota

### 2.1. Gut Microbial Structure and IBS Pathological State

In recent years, advancements in technology such as whole-genome shotgun sequencing and multi-omics analysis have unveiled specific dysbiosis patterns unique to IBS. Comparative analysis of 16S rRNA gene sequencing revealed structural differences between the fecal and mucosal microbiota of IBS patients, with a significant correlation between the two. Fecal samples exhibited higher levels of Firmicutes and Actinobacteria, while the mucosal microbiota was enriched in Bacteroidetes and Proteobacteria. Overall microbial diversity in stool samples from IBS patients is lower compared to healthy individuals [15]. Furthermore, IBS patients demonstrated a notable increase in the Firmicutes/Bacteroidetes ratio, with significant variations observed in families such as *Ruminococcaceae* and *Clostridiaceae*, as well as genera including *Ruminococcus*, *Bifidobacterium*, and *Faecalibacterium* spp. [16,17]. The specific mechanism of intestinal flora affecting IBS is shown in Table 1.

Studies have indicated a downregulation of probiotic colonization in IBS patients, particularly involving *Lactobacillus*, *Bifidobacterium*, and *Faecalibacterium prausnitzii* [18]. Additionally, patients with IBS exhibited elevated levels of *E. coli* and *Enterobacter* [19]. The composition and abundance of intestinal microbiota were found to be associated with various symptomatic parameters of IBS. For instance, the *Clostridiales*-rich intestinal type displayed longer runs compared to the *Prevotella-* and *Bacteroides*-rich type. Moreover, IBS-D patients showed higher levels of *Bacteroides* compared to IBS-C patients. As the severity of IBS symptoms increased, there was a gradual decrease in the abundance of *Prevotella* and an increase in *Bacteroides* [20]. Notably, the frequency and characteristics of fecal matter in IBS-D patients are negatively correlated with the abundance of *Lactobacillus* and *Bifidobacteria* [21]. In addition to changes in the large intestine, IBS patients experienced alterations in the small intestinal microbiota, including small intestinal bacterial overgrowth and a notable increase in *Pseudomonas aeruginosa* [22,23]. Small intestinal bacterial overgrowth in IBS patients is characterized by an overgrowth of colonic bacteria such as *Klebsiella*, *Enterococcus*, and *Escherichia* [24]. Furthermore, post-infectious IBS (PI-IBS) can develop in 10–30% of patients following acute infectious diarrhea, with gut microbial disorders playing a role in the pathogenesis of PI-IBS [25]. Studies on PI-IBS mice revealed reduced levels of colon *Lactobacillaceae* and *Alistipes*, along with indications of low-grade inflammation and increased intestinal permeability [26,27], consistent with decreased *Alistipes* taxon abundance in PI-IBS patients [28]. However, due to limitations in clinical sample size, it remained challenging to ascertain whether specific dysbiosis occurred universally in all individuals with IBS and whether this dysregulation is a cause or consequence of the condition.

Numerous studies have investigated the impact of probiotic supplementation on IBS to better understand the role of gut microbes in this condition. Meta-analyses and systematic reviews of fifty-three RCTs of probiotics involving 5545 patients have highlighted that specific combinations of probiotic or particular species and strains are more effective than a placebo in alleviating overall IBS symptoms and abdominal pain [29]. In the case of IBS-C patients, they enhance stool consistency, increase levels of *Bifidobacteria* and *Lactobacilli* in stool, and demonstrate a favorable safety profile [30]. Subsequent research endeavors have aimed to identify which strains of *Bifidobacteria* and *Lactobacilli* are most beneficial for improving IBS symptoms. Notably, one revealed that *B.* BB536 and *L.* HN001 could effectively reduce abdominal pain and bloating while restoring intestinal barrier function in individuals with IBS [31]. Furthermore, *L. paracasei* HA-196 and *B. longum* R0175 have been found to alleviate gastrointestinal symptoms and enhance mental health in IBS patients [32]. Additionally, *B. coagulans* had shown promising results in alleviating overall IBS symptoms, *L. plantarum* has demonstrated efficacy in enhancing the quality of life for IBS patients, and *L. acidophilus* exhibited the lowest incidence of adverse events during treatment. *Lactobacillus acidophilus* NCFM can modify the expression of pain-associated receptors, such as μ-opioid and cannabinoid receptors, in the GI tract in mice and humans, thereby improving the symptoms of abdominal pain [33]. A meta-analysis of 82 trials containing data from more than 10,000 patients has demonstrated that potentially effective probiotic compositions (mainly including *Bifidobacterium*, *Lactobacillus*, *Enterococcus faecalis*, *Streptococcu*) for improving IBS symptoms had a better improvement effect than single probiotics [34]. However, evaluating efficacy remains challenging due to limited sample sizes, inadequate study designs, and the inconsistent standardization of utilized probiotic strains. The integration of multi-omics techniques is essential to unravel the impact of gut microbes on gut motility and health in animal models of IBS, offering novel insights into the pathophysiology underlying IBS. Current evidence suggests that microbial alterations can be crucial for enhancing overall symptom management in IBS patients. However, the specific probiotic combinations, species, or strains that are effective for IBS treatment remain unclear [29], warranting further investigation into the underlying mechanisms.

Moreover, FMT has emerged as an effective therapy for IBS patients, particularly when utilizing carefully selected donors with specific beneficial microbial profiles. A recent study showed that FMT can reduce symptoms in some IBS patients, particularly those with bloating, with significant improvements in both the short and long term [35]. However, the effect may vary depending on the individual microbiome. A five-year follow-up study at Shanghai No. 10 Hospital showed significant improvements in IBS severity scores, bloating, and abdominal pain after FMT treatment [36]. A meta-analysis of 472 patients with IBS from seven randomized controlled trials showed that FMT significantly relieved patients’ discomfort [37]. FMT has shown dose-dependent improvements in gastrointestinal symptoms [38]. Constipation patients have been found to exhibit higher levels of Bacteroides in the colonic mucosal microbiota, and FMT from constipated donors has been shown to delay gastrointestinal transit time in mice [39]. Additionally, an increased abundance of Methanobrevibacter has been observed in the feces of constipated donors, aligning with heightened methane production in individuals with chronic constipation, which is believed to slow intestinal motility [40]. Clinical studies have demonstrated significant improvement in clinical symptoms in patients with IBS-D by decreasing levels of *Escherichia coli*, increasing levels of *Bifidobacteria*, and decreasing levels of isovaleric acid and valeric acid following treatment of subjects with FMT [41]. These findings suggested that alterations in the gut microbiota may underpin a more precise understanding of the pathogenesis and management of IBS symptoms. Overall, FMT was well tolerated and the incidence of adverse reactions was low in IBS patients. There are several potential risks associated with FMT, including the possible transmission of undiagnosed pathogens from the donor, immune responses in some recipients, and unpredictable outcomes due to differences in individual microbiome characteristics. Other issues include immunosuppressive therapy, drug use, and changes in the recipient microbiome, which may affect the success and stability of FMT therapy [42]. Strict preoperative screening of donors and adequate preoperative preparation of patients can make FMT a relatively safe treatment. Although FMT has provided significant therapeutic effects for some patients in the short term, its long-term effects, potential risks, and indications still require further scientific research for confirmation. In clinical practice, FMT should be carefully used under the guidance of doctors according to the specific conditions and treatment needs of patients, while a large number of clinical observations and long-term follow-up are needed to evaluate its role and impact, so as to conduct a comprehensive risk assessment. Further research is needed to identify the beneficial microbiota and the mechanisms involved in ideally transferring a range of well-defined “therapeutic” microbiota and avoiding the risk of introducing potential pathogens.

### 2.2. Gut Microbial Metabolites and IBS Pathological State

In addition to the gut microbiota itself, the gut microbiota plays a critical role in preserving the healthy gut of the host through gut microbiota-derived metabolites, as shown in Table 2.

Short-chain fatty acids (SCFAs), including acetate, propionate, and butyrate, are primarily synthesized by colonic microorganisms through the fermentation of dietary starch. They exert both inhibitory and stimulating effects on colonic motility, and their metabolic dysregulation is associated with the development of IBS [43]. Meta-analyses have indicated that propionate and butyrate can serve as biomarkers for diagnosing IBS, with decreased levels of fecal propionate and butyrate observed in IBS-C patients and increased butyrate concentrations in IBS-D patients [44]. Increased intestinal motility and colon fermentation stimulation may lead to increased levels of SCFAs in patients with IBS-D, slower intestinal absorption, and accumulation in the stool. In patients with IBS-C, reduced acetate concentrations may be associated with slower bowel movement, which affects the fermentation process and SCFA production in the gut [45]. SCFAs stimulate the release of 5-hydroxytryptamine (5-HT) from gut enterochromaffin cells, promoting the release of acetylcholine (Ach) from the colonic muscle plexus and accelerating colonic transit [46]. In PI-IBS, there is a notable reduction in SCFA production and an exacerbation of intestinal inflammation, leading to decreased tryptophan production in the later stages of infection, significant decline in the indole pathway, and diminished aryl hydrocarbon receptor (AhR) activity. These alterations manifested as colonic hypersensitivity and anxiety-like behavior [26]. The vagus nerve’s potential involvement in CNS–microbiota communication can influence emotional and neurobehavioral disorders in humans, contributing to the onset and progression of IBS. SCFAs can directly activate free fatty acid receptors in various intestinal cells, subsequently entering the brain via the vagus nerve to regulate intestinal physiological functions [47].

Tryptophan (Trp), an essential amino acid, plays a crucial role in host–microbiome crosstalk through its metabolism, thereby maintaining gastrointestinal function [48]. The administration of *B. infantis* strain 35,624 increased tryptophan blood levels in depression-like rats [49]. Approximately 1% of Trp is converted to 5-HT in humans, with the majority undergoing catabolism to kynurenine and quinolinic acid by indoleamine 2,3-dioxygenase (IDO1). Certain bacteria, such as *Corynebacterium* spp., *Streptococcus* spp., and *Enterococcus* spp., can also directly produce 5-HT [50,51], which plays a pivotal role in gastrointestinal motility by stimulating intrinsic sensory neurons to initiate the peristaltic reflex [52]. Elevated blood levels of 5-HT have been observed in patients with IBS-D, correlating with the degree of visceral pain and hypersensitivity [53]. Moreover, patients with IBS-D exhibit abnormal intestinal mucosal 5-HT metabolism, characterized by increased colonic 5-HT levels, reduced expression of the serotonin transporter (SERT), and visceral hypersensitivity [54]. Decreased *Helicobacter* abundance and concentration of *Butyricimonas* improved symptoms and serotonin levels in the colons of IBS rats [55]. Under normal circumstances, the gut microbiota members established stable microbial complexes through 5-HT-mediated quorum sensing interactions, and disruption of the 5-HT pathway may contribute to the microecological dysregulation observed in IBS. The kynurenine pathway also plays a multifaceted role in IBS pathophysiology. Kynurenic acid exerts regulatory functions in the central nervous system (CNS) and gastrointestinal tract via the G-protein-coupled receptor GPR35, potentially alleviating inflammatory pain in IBS [56,57]. Reduced levels of kynurenic acid in the intestinal mucosa have been observed in patients with IBS, correlating significantly with gastrointestinal function [58]. Some intestinal microorganisms, including *E. coli*, *Achromobacter liquefaciens*, and *Bacteroides* spp., can convert tryptophan to indole, affecting host intestinal function through AhR [48]. The activation of AhR promotes the production of anti-inflammatory cytokines such as IL-10, IL-22, and Foxp3 while inhibiting pro-inflammatory cytokines such as IL-6, IL-12, and IFN-γ, thereby ameliorating intestinal inflammation and promoting mucosal barrier repair [59]. *L. reuteri*, *L. bulgaricus*, *Alistipes onderdonkii*, or their tryptophan metabolites, including indole-3-ethanol, indole-3-pyruvate, indole-3-aldehyde, and 5-hydroxyindole-3-acetic acid, can regulate intestinal inflammation via AhR-dependent mechanisms [60,61]. The colonization of germ-free mice with *Ruminococcus gnavus* induces the catabolism of dietary phenylalanine and tryptophan to produce phenylethylamine and tryptamine, stimulating 5-HT biosynthesis in ECs by activating trace amine-associated receptor 1 (TAAR1), thereby inducing symptoms of IBS-D [57]. Tryptamine produced by *Bacteroides thetaiotaomicron* enhances ion flux and intestinal fluid secretion in the colonic epithelium of germ-free mice, thereby accelerating gastrointestinal transit [62].

Bile acids (BAs) are synthesized from cholesterol in the liver and stored in the gallbladder. Upon food consumption, BAs are released into the gastrointestinal tract and undergo further modifications by the gut microbiota to form secondary BAs with diverse biological activities [63]. Among these, deoxycholic acid (DCA) and lithocholic acid (LCA) stand out as significant secondary BAs produced through microbial biotransformation in the colon, known to enhance gastrointestinal motility by activating TGR [64,65]. Research has shown that the levels of unbound primary BAs are significantly increased in the stool samples of IBS-D patients, while the BA content in the stool samples of IBS-C patients is significantly decreased, especially with reduced levels of primary BAs (such as cholic acid) and LCA, which can cause bowel movement to slow down and lead to constipation [8]. Certain Clostridium species possess 7α-dehydroxylase activity, converting cholic acid to DCA [66], and colonization with spore-forming microbes containing *Clostridium* has shown promise in ameliorating gastrointestinal dyskinesia associated with germ-free conditions. Additionally, this colonization activates IPAN in the colonic muscle cluster, further supporting gastrointestinal function [67]. In patients with IBS-D, fecal BAss enriched with *C. scindens* have been positively correlated with serum levels of 7α-hydroxy-4-cholestene-3-one (C4) [68]. Studies, through meta-analysis and systematic review, have revealed that bile acid malabsorption is present in 28% of IBS-D patients, with BA sequestrants such as coleylenide showing significant therapeutic benefits [69,70].

Histamine, a biogenic amine, is implicated in IBS pathogenesis by modulating gastrointestinal motility, mucosal permeability, and ion secretion [71]. Resident intestinal microbes, including *E. coli* and *Morganella morganii*, are identified as histamine-producing strains [72], with some Gram-negative bacteria capable of histidine decarboxylation (HDC) to convert histidine to histamine [73]. Supplementation with *L. reuteri* facilitates the conversion of L-histidine to histamine in the gut, activating the histamine H2 receptor (H2R) to mitigate intestinal inflammation [74]. Elevated levels of histamine have been observed in the colon samples of IBS patients compared to healthy individuals, along with significantly increased mRNA levels of H1R and H2R in colon tissues [75]. The released supernatant can sensitize the mouse transient receptor potential vanilloid 1 (TRPV1) through the H1R [76]. Therefore, dysbiosis involving histamine or HDC-secreting bacteria may play a role in the onset and progression of IBS. Moreover, the gut microbiota contributes to the synthesis of various neurotransmitters such as norepinephrine (NA), dopamine (DA), gamma-aminobutyric acid (GABA), histamine, and ACh, impacting both the psychological and physiological aspects of IBS [77,78]. NA, a catecholamine neurotransmitter, plays essential roles in physiological processes like the cardiovascular and endocrine systems. Inhibitors targeting the reuptake of 5-HT and NA have shown therapeutic benefits in IBS-D patients and animal models [79]. DA, a precursor to NA and adrenaline, is associated with chronic pain perception and anxiety. It also regulates intestinal contractility, with DA-D2 receptor antagonists demonstrating efficacy in reducing visceral hypersensitivity and enhancing colonic permeability in IBS rat models [80].

Quercetin, a flavonoid commonly present in various vegetables, fruits, and grains, is predominantly synthesized by gut bacteria within the host, including *Fusobacteria* [81]. Its beneficial effects includ the modulation of mAChR signaling to alleviate loperamide-induced chronic constipation [82]. Moreover, in PI-IBS mice, quercetin has demonstrated the ability to ameliorate visceral hypersensitivity by reducing EC density, decreasing 5-HT levels in the colon, and regulating tryptophan hydroxylase (TPH) expression [83]. A systematic review has shown that quercetin improves clinical symptoms and mood disorders in IBS-D patients by regulating the immune response, oxidative stress, and balancing neurotransmitters [84].

Putrescine and cadaverine produced by intestinal bacteria act on intestinal chemical sensors, regulate intestinal peristalsis, and are considered potential biomarkers of IBS-D disease. Intestinal proteases not only facilitate digestion but also mediate immune signaling. Elevated intestinal proteolytic activity driven by host serine proteases can disrupt intestinal barrier integrity, leading to visceral hypersensitivity. High intestinal proteolytic enzyme activity has been observed in PI-IBS rats. Furthermore, β-glucuronidase produced by *Alistipes* counteracts protease activity by modulating bilirubin production [28].

Certain pathogens have the capacity to secrete pore-forming toxins or N-formylated peptides, contributing to increased visceral sensitivity by triggering the release of inflammatory mediators, depolarizing nociceptor neurons, and compromising epithelial integrity [85].

Overall, the dysregulation of metabolic pathways and microbiota composition contributes to the development of different IBS subtypes. IBS-D is characterized by diarrhea, disruptions in BA metabolism, and reduced SCFA production, leading to gastrointestinal motility dysfunction and hypersensitivity. IBS-C, primarily characterized by constipation or infrequent bowel movements, is associated with a decrease in SCFAs and a reduction in the beneficial bacteria that produce these metabolites. PI-IBS typically occurs following infection and involves disruptions in Trp metabolism, leading to decreased serotonin levels and increased visceral hypersensitivity. Metabolites, including SCFAs, BAs, serotonin, and kynurenine, play key roles in IBS pathogenesis by regulating gastrointestinal motility, inflammatory responses, and neurotransmitter production. Understanding these metabolic mechanisms provides strong evidence for therapies that target microbiome balance to reduce symptoms of IBS.

### 2.3. Cellular Components of Gut Microbes and the Pathological State of IBS

Intestinal bacterial components play a significant role in influencing symptoms related to IBS through signaling interactions among gastrointestinal cells. For instance, certain bacterial cell wall components such as lipopeptides, peptidoglycans (PGNs), flagellin, and LPS regulate intestinal motility by binding to specific subtypes of Toll-like receptors (TLRs) present in intestinal cells. Among these TLRs, TLR2, and TLR4 are key regulators of the enteric nervous system (ENS) and intestinal motility. TLR2 detects lipopeptides and PGNs, while TLR4 recognizes LPS [86].

TLR2 is expressed in various cells within the intestinal wall, including intestinal neurons, glial cells, and smooth muscle cells. Studies on TLR2-deficient mice (TLR2^−/−^) have shown impaired ENS structure, reduced glial cell line-derived neurotrophic factor (GDNF) synthesis, abnormal mucosal secretion, and disruptions in intestinal motility, similar to observations in wild-type mice with depleted intestinal microbiota. However, GDNF administration effectively reverses these ENS defects and gastrointestinal motility issues, underscoring the pivotal role of the intestinal microbiota-TLR2-GDNF axis in ENS function and gastrointestinal motility. Lipopeptides, prominent components of Gram-positive intestinal bacteria, have exhibited anti-inflammatory effects by activating TLR2 and modulating gastrointestinal motility independently of MyD88. Bacterial lipopeptides are recognized as key players in IBS [82]. In addition, the outer membrane protein Amuc_1100 from *A. muciniphila* enhances gastrointestinal motor function through TLR2 signaling [87]. *B. thetaiotaomicron* has been found to increase intestinal neuron populations, boost Ach and SP secretion, enhance colon motility, and increase TLR2 expression in the colons of germ-free mice [88]. *C. butyricum* promotes the release of auxin-releasing peptide and SP, regulates cells of Cajal viability in a TLR2-dependent manner, and supports gastrointestinal motility [89]. Unfortunately, it has not yet been determined which bacterial components are responsible.

Similarly, TLR4^−/−^ has exhibited aberrant ENS development, decreased neuronal nitric oxide synthase in the colonic muscle cluster, and consequent intestinal motility disorders [90]. Studies have indicated significantly increased TLR4 expression in IBS-D patients [91] and elevated TLR2 expression in the monocytes of IBS patients [88], correlating with gastrointestinal and psychological symptoms in affected individuals. Serum levels of LPS and anti-flagellin antibodies are found to be elevated in patients with IBS-D, showing a correlation with anxiety scores [92]. LPS is known to inhibit the contraction of intestinal smooth muscles by interacting with TLR4 and IKK-β kinase complexes [89]. Both pathogenic *Shigella* and non-pathogenic *E. coli K-2* produce LPS, impacting the peristalsis of human colon smooth muscle cells through the activation of TLR receptors [93]. In individuals with IBS, LPS can modulate TLR4 signal transduction in neurons, thereby influencing gastrointestinal motility [94]. Myographic macrophages (MMs), a specific subtype of macrophages, release bone morphogenetic protein 2 (BMP2) upon stimulation by LPS, activating BMP receptors expressed by intestinal neurons to regulate gastrointestinal motility [95]. Conversely, heightened levels of LPS can trigger an inflammatory response in the gastrointestinal tract, leading to smooth muscle dysfunction and the impairment of gastrointestinal motility [96]. The inhibitory impact of LPS on gastrointestinal motility is contingent upon time. Early exposure to LPS induces cyclooxygenase-2 (COX-2) production of PGE2, which hampers smooth muscle contraction. In contrast, late-stage exposure results in the upregulated expression of inducible nitric oxide synthase (iNOS), leading to nitric oxide (NO) production and promoting contraction [97]. Hence, the influence of LPS dose and type on intestinal motility warrants further investigation. Similarly, TLR5 and TLR9 are upregulated in patients with IBS, indicating a potential connection between the dysregulation of intestinal symbiota and the immune response in IBS patients [98]. Flagellin, produced by intestinal bacteria *Lachnospiraceae*, acts as a TLR5 agonist [95], capable of activating innate immunity in the intestine [96]. Studies have shown that flagellin activates innate and adaptive immunity in patients with inflammatory bowel disease, with significantly higher flagellin antibodies in the IBS group compared to the normal healthy group [99].How do gut microbiota, metabolites, and cellular components affect IBS as shown in Figure 1.

**Figure 1 biomolecules-14-01643-f001:**
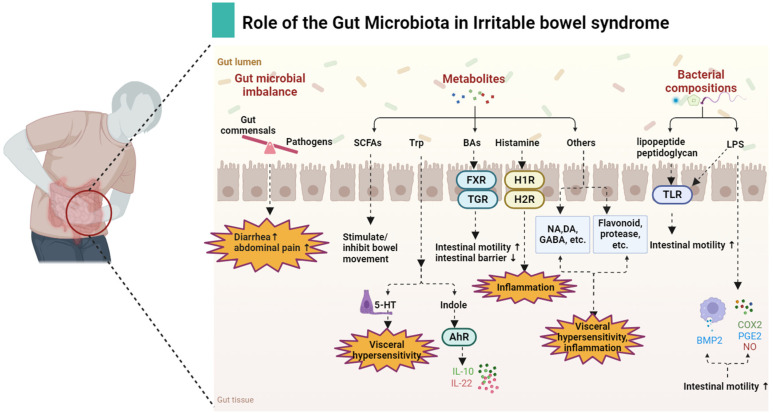
Interactions and connections between the gut microbiota, metabolites, and cellular components with IBS. Imbalance in the intestinal gut microbiota can lead to symptoms such as diarrhea and abdominal pain. Gut microbiota metabolites, such as SCFAs, can directly affect intestinal movement, BAs affect intestinal movement and intestinal barrier function via the FXR/TGR pathway, Trp affects visceral hypersensitivity and intestinal inflammation through conversion to 5-HT and indol, and histamine affects intestinal inflammation through its receptors. Neurotransmitters (NA, DA, GABA), flavonoids, and proteases can also influence visceral hypersensitivity and inflammation. Gut microbiota components, lipopeptide, peptidoglycans, and LPS act on the TLR pathway or stimulate intestinal macrophages to secrete BMP2 and inflammatory factors (COX2, PGE2, NO) to affect intestinal movement.

## 3. Diet-Mediated Gut Microbiome Changes and IBS

Diet plays a crucial role in impacting both mental health and gastrointestinal symptoms in individuals with IBS. Self-reported triggers for IBS symptoms often involve foods rich in carbohydrates, fats, histamines, and amines, all of which have been linked to the severity of symptoms and overall quality of life [100].

IBS symptoms are often exacerbated by malabsorbable fermentable oligosaccharides, disaccharides, monosaccharides, and polyols (FODMAPs) as well as insoluble fibers, the FODMAP classification of common foods is shown in Figure 2. These components increase osmotic pressure in the large intestine, providing a substrate for bacterial fermentation in the distal small intestine and colon. This process leads to gas production, triggering bloating, abdominal pain, or discomfort [101]. Diet significantly influences the host’s mood, behavior, and gastrointestinal motility, mainly through the gut microbiota.

Adopting a low-FODMAP and -insoluble fiber diet has been shown to reduce symptoms and enhance the quality of life for individuals with IBS. These dietary changes also positively influence the gut microbiota, helping to improve abnormalities in gastrointestinal and endocrine cells [102]. Multiple studies have shown that low-FODMAP diets and low-carb diets reduce fermentation and gas production and regulate the gut microbiota. These dietary interventions have shown significant IBS symptom improvement efficacy both in the short term and at follow-up, while also enabling personalized treatment strategies based on triggers [103,104]. Specifically, the low-FODMAP diet has been found to decrease the abundance of the *B. adolescentis* strain, known to disrupt intestinal epithelial TJ integrity and intestinal barrier function [101]. In comparison to healthy individuals, those with IBS who adhere to a low-FODMAP diet exhibit lower fecal *Bifidobacterium* abundance post intervention [105,106]. Controlled clinical trials have further demonstrated that a low-FODMAP diet intervention in IBS patients reduced the abundance of gas-producing bacteria, increased *Actinobacteria* levels, and decreased histamine release [107]. The dysbiosis of gut microbes resulting from a high-fructose diet can lead to barrier damage and low-grade gut inflammation [108]. The duration of a low-FODMAP diet is generally 3 to 6 weeks, and high-FODMAP foods should be gradually reintroduced after the short-term adoption of a low-FODMAP diet because long-term low-FODMAP diets may alter the gut microbiome and even reduce *Akkermansia* [109].

**Figure 2 biomolecules-14-01643-f002:**
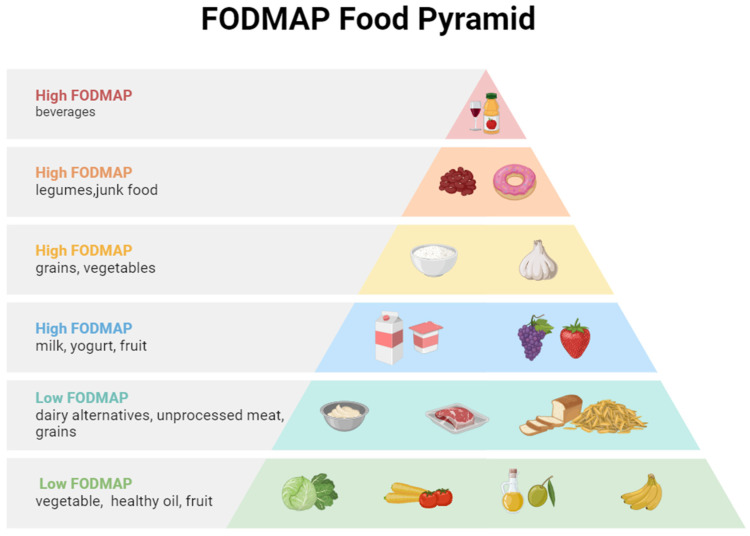
FODMAP classification of common foods.

While junk food and legumes are both rich in FODMAPs that can cause gastrointestinal distress, they contain different types. Legumes are mainly composed of oligosaccharides (especially galacto-oligosaccharides), such as fermentable carbohydrates, which can lead to the production of gas in the gastrointestinal tract. The hypothesis that junk food causes bloating and abdominal pain includes factors such as increased fat content that alters bowel movements, excess salt, spices and preservatives, and a lack of fiber that is good for human health.

Moreover, research has indicated that dairy products play a significant role in triggering or exacerbating IBS-D symptoms. A high-lactose diet can induce diarrhea in mice rapidly, increase fecal water content, shorten fecal transport time, and elevate the number of mast cells in the colonic mucosa through glycosylation reactions, ultimately heightening visceral sensitivity [110].

“Healthy” dietary patterns containing fish or omega-3 FA dietary supplements have been found to be associated with a reduced risk of depression associated with IBS. However, a decrease in fish consumption or omega-3 fatty acids may induce gut inflammation by impacting the microbiome. Omega-3 fatty acid deficiency results in an increase in the Firmicutes/Bacteroidetes ratio, and with omega-3 fatty acid supplementation, the levels of *Bifidobacterium* and *Lactobacillus* increase, and the abundance of *Anaeroplasma*, *Clostridium*, and *Peptostreptococcaceae* decreases [111,112]. Studies have shown that supplementation with the omega-3 fatty acids eicosapentaenoic acid/docosahexaenoic acid can ameliorate intestinal microbiota imbalances in rats exposed to early life stress, such as maternal separation [113].

Micronutrients such as zinc, magnesium, selenium, iron, vitamins, and folate can impact IBS symptoms by influencing neurotransmitter production and activity, inflammation, and oxidative stress responses [114]. Mice on a magnesium-deficient diet exhibited disrupted gut microbiota and increased depression-like behavior [115]. Vitamin D supplementation has been found to balance the composition of gut microbiota and reduce the abundance of *Pseudomonas* spp., *Escherichia*, and *Shigella* spp. [116]. Iron, an essential trace element for most intestinal bacteria, plays a crucial role, as dietary iron restriction in mice leads to increased levels of *Lactobacilli* and *Enterococci* [117,118] while reducing the presence of butyrate-producing *Roseburia* spp. and *E. rectale* [119]. Additionally, iron-rich foods such as meat and fish may influence gut microbiota composition, potentially exacerbating this imbalance. Diets rich in fruits and vegetables containing micronutrients like vitamin C, polyphenols, and flavonoids have demonstrated antidepressant/anxiolytic effects, anti-inflammatory properties, and gastrointestinal mucosal protection [120]. Therefore, future comprehensive dietary studies should explore the correlation between micronutrient intake, gut microbiota composition, and IBS symptoms.

As shown in Figure 3, dietary choices play a significant role in shaping the gut microbiome structure. Novel dietary interventions are being developed to reshape the microbiome, potentially alleviating and preventing low-grade intestinal inflammation and gastrointestinal symptoms in individuals with IBS. Changes in the gut microbiota mediated by drug and food homology are also worth exploring in the prevention and treatment of IBS [121]. Additionally, the impact of diet-induced changes in microbiome structure on local intestinal tissue function cannot be fully elucidated using monolayer cell cultures or incomplete organoids. More advanced platforms are needed to understand how diet-induced modifications in microbiome composition regulate intestinal gene expression and functional changes.

**Figure 3 biomolecules-14-01643-f003:**
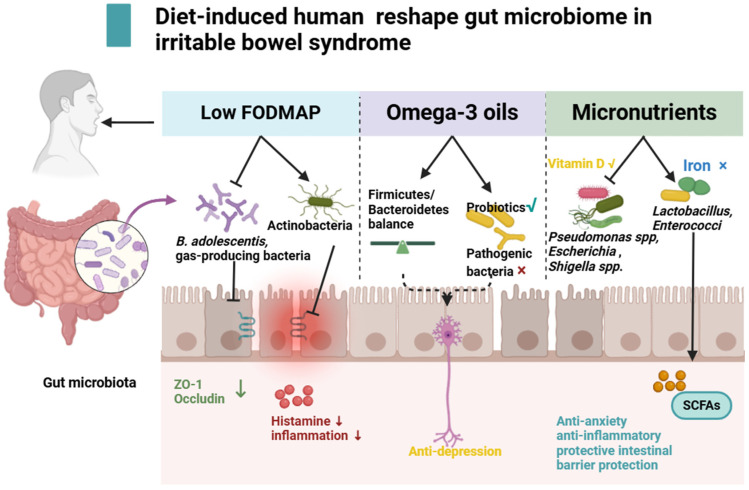
The effect of diet on IBS by acting on the gut microbiota and its metabolites. Low FODMAP promotes Actinobacteria and inhibits *B. adolescentis* and gas-producing bacteria to improve intestinal barrier dysfunction and inhibit inflammation. A fish and omega-3 oil diet, via balancing intestinal F/B, increases probiotic abundance and plays an antidepressant role. Vitamin D supplementation can reduce the abundance of *Pseudomonas* spp., *Escherichia*, and *Shigella* spp. Iron restriction increases levels of *Lactobacillus* and *Enterococcus*, promotes SCFA secretion and anti-anxiety and -inflammation effects, and improves intestinal microenvironment.

## 4. “Brain-Gut Axis”—The Pathway Through Which Gut Microbes Influence IBS

The brain and gut have a profound connection when it comes to health and disease. Gut microbes play a crucial role in their communication, primarily through the synthesis of various neuroactive molecules that impact the CNS and neuroendocrine responses. Conversely, signals from the CNS and neuroendocrine responses can influence the composition of the gut microbiota.

With the introduction of updated diagnostic criteria (Rome IV criteria), a new umbrella term, DGBI, has been coined to encompass the multiple pathological processes associated with IBS linked to deficits in brain–gut axis communication. Often, patients with gut–brain interaction disorders have functional dyspepsia (FD) or IBS [122]. Up to one-third of people with IBS also experience anxiety or depression, and psychological comorbidity seems to be more important for long-term quality of life [123]. Clinical observations have indicated that a majority of IBS patients exhibit heightened levels of trait anxiety and neuroticism, or meet the diagnostic criteria for anxiety disorders [124,125]. Additionally, research has suggested that gut microbes play a role in the development of the CNS, particularly in systems related to stress response, anxiety, depression, and memory [117,118]. Communication between the gut microbiota and the central and peripheral nervous systems involves activations such as the vagus nerve, the stimulation of endocrine cells, immune-mediated signaling, and the direct transfer of gut microbial metabolites to the brain.

### 4.1. Vagus Nerve Activation

The vagus nerve serves as the primary anatomical link between the gut and the brain, functioning as a defined signaling pathway that influences host behaviors like feeding, anxiety, depression, and social interactions. By intercepting signals from the gut microbiota, the vagus nerve transmits these messages to the CNS, impacting the host’s nervous and digestive systems [126]. Patients with IBS often exhibit autonomic abnormalities characterized by reduced vagal tone, intestinal barrier damage, and disorders in intestinal microecology [127]. Although the afferent fibers of the vagus nerve are distributed in the intestinal wall without direct contact with microorganisms in the gut lumen, they can sense bacterial metabolites by absorbing across the epithelium or indirectly perceiving microbiota signals through signals transmitted by other epithelial cells. Different SCFAs activate vagal afferent fibers through various mechanisms. For instance, butyrate exerts remote control over the brain by activating vagal afferent neurons [128]. Intestinal epithelial cells detect microbial signals via TLRs, indirectly influencing vagal afferent fibers and thereby regulating gastrointestinal movements and secretions.

Intestinal epithelial cells detect microbial signals through TLRs and regulate gastrointestinal movement and secretion through indirect effects on vagal afferent fibers (See Section 2.3 for details). *Campylobacter jejuni* can activate the vagus nerve, resulting in increased expression of c-Fos, a neuronal activation marker in the sensory ganglia and vagus nerve [129], indirectly emphasizing the microbial activation of the afferent fibers of the vagus nerve. A duodenal injection of *L. johnsonii* can enhance vagal nerve activity [130]. Mice with chronic exposure to *L. rhamnosus* JB-1 showed altered GABA brain expression, and all of these animals exhibited stress-induced anxiety and depression behaviors [131]. In summary, the vagus nerve mediates the influence of gut microbes on mood and gastrointestinal movement, so drugs that target the microbiome and vagus tone regulation can help treat IBS.

Single-cell sequencing revealed that the ENS contains multiple types of neurons that sense and respond to chemical and mechanical stimuli, transmit signals, induce other types of secreted cells, control blood flow, and regulate gastrointestinal muscle contraction and relaxation. These neurons in the intestinal wall respond to gut microbes and their metabolites in the lumen and directly or indirectly regulate gastrointestinal motility. Intestinal microbial derivatives bind to luminal cells, ECs, and L cells, activate downstream signals, and activate receptors on intestinal neurons to regulate gastrointestinal motility [132]. At the same time, studies have shown that the interaction between intestinal neurons and intestinal microbes can promote the survival of intestinal neurons and gastrointestinal motility in mice. For example, nitrogen energy neurons in germ-free mice are significantly reduced, and intestinal peristalsis is delayed compared with conventional mice [94]. Compared with conventional rats, germ-free rats also showed significant delays in intestinal transport and slight intestine contraction, which were partially reversed by colonizing *L. acidophilus* A10 and *B. bifidum* B11 [133].

### 4.2. Stimulation of Endocrine Cells

ECs serve as the primary endocrine cells in the intestine, playing a crucial role in intestinal signaling and accounting for 90% of 5-HT secretion in the human body. Beyond its function as a neurotransmitter, 5-HT acts as a vital regulator for the gastrointestinal tract and other organ systems, providing valuable insights into the intricate interactions within the brain–gut–microbiome axis. TPH acts as a pivotal enzyme in 5-HT biosynthesis, and ECs express two TPH subtypes (TPH-1, TPH-2) [134,135]. Research indicates that gut microbes can directly transmit signals to ECs, leading to the induction of TPH-1 and TPH-2 transcription and promoting 5-HT synthesis. Certain resident microorganisms, such as *Streptococcus* spp., *Enterococcus* spp., and *Clostridium ramosum*, are also capable of directly producing 5-HT [136]. A comparative analysis has revealed a significant reduction in 5-HT content in the colon and cecum of germ-free animals when compared to conventional animals, suggesting a critical role of gut microbes in regulating host 5-HT levels [137]. Germ-free mice that received FMTs exhibited a marked increase in 5-HT-positive endocrine cells and myometrial-positive macrophages in the upper digestive tract and colon, along with accelerated bowel functioning [138]. Additionally, ECs responded to intestinal microbial metabolites (such as SCFAs and aromatic amino acid metabolites) by promoting TPH1 transcription and increasing mucosal 5-HT concentration [139]. Fermented supernatants from *Bacillus subtilis*, *Enterococcus faecium*, and *Enterococcus faecalis* have been shown to upregulate SERT expression in intestinal epithelial cells and intestinal tissues of PI-IBS rats [140]. Furthermore, certain gut microbes possess tryptophan enzymes that convert tryptophan to indole, thereby limiting the availability of tryptophan to the host. For instance, *B. uniformis*, *B. fragilis*, and *B. caccae* contain tryptophanase, which deplete 5-HT in the hippocampus, potentially triggering symptoms of depression [141]. Studies have indicated that *L. plantarum* PS128 reduces anxiety-like behavior by increasing dopamine and 5-HT concentrations in the brain’s striatum in mice [142]. Collectively, these experiments highlight the potential of intestinal microorganisms to activate intestinal endocrine cells, regulate host 5-HT biosynthesis, and contribute to IBS symptoms. Neurotransmitters play a significant role in regulating the development and function of the ENS and CNS [143,144]. However, most neurotransmitters have limited half-lives and an uncertain ability to cross the blood–brain barrier, raising questions about their capacity to reach specific locations within the CNS at sufficient concentrations.

Intestinal endocrine cells are capable of sensing microbial metabolites and secreting various hormones, including glucagon-like peptide 1 (GLP-1) and peptide YY (PYY), which play a role in IBS-related intestinal endocrine functions [145]. L cells in the intestine secrete GLP-1 in response to stimulation by SCFAs, BAs, and other bacterial derivatives, serving as a signal transducer in the brain–gut–microbiome axis [146]. It has been observed to enhance colonic transport, and a reduced expression of circulating GLP-1 and mucosal GLP-1 receptors has been associated with disease severity in patients with IBS-C [134]. The intestinal microbiota has the capacity to alter gastrointestinal motility while inhibiting GLP-1 receptor expression in gastrointestinal spinal cord nerve cells [135]. Similarly, PYY primarily stimulates the secretion and production of intestinal L cells by microorganisms, influencing the release of 5-HT by ECs to regulate gastrointestinal peristalsis and secretion [136]. Clinical studies have demonstrated a significant reduction in the concentration of PYY and the number of PYY cells in the colons of patients with IBS, potentially contributing to intestinal motility disorders and visceral hypersensitivity in these individuals [137].

### 4.3. Immune Activation

The lamina propria of the intestine is densely populated with macrophages, dendritic cells (DCs), and lymphocytes, which serve as the initial responders to microbial signals. Upon sensing these signals, they present antigens to T cells, leading to the differentiation and activation of various T cell subpopulations such as Th1, Th2, Th17, or Treg. The gut microbiota plays a pivotal role in shaping the development and functionality of the neuroimmune system [147]. In patients with IBS, abnormal neuro-immune interactions have been observed, characterized by an increased presence of mast cells and monocytes in the intestinal mucosa and blood. Additionally, the direct activation of pain-transmitting nerves by mast cell-releasing mediators, elevated T cell numbers in the intestinal mucosa, and alterations in B-cell activity and antibody production had been documented [148,149]. Notably, individuals with PI-IBS exhibit significantly higher quantities of intestinal chromaffin cells, mast cells, and lamina propria T lymphocytes [150]. Moreover, the abundance of intestinal immune cells in IBS have been linked to gender, with female IBS patients showing higher levels of mast cells but lower counts of CD3+ and CD8+ T cells [151].

Furthermore, the diminished presence of *A. muciniphila* in the guts of IBS patients are been associated with an upregulation of the Treg cell response, leading to improvements in neuroinflammation [145]. Following supplementation with *B. infantis* EVC001, there is an increase in the content of indolyl lactic acid in feces, which was found to be involved in regulating the differentiation of Th2 and Th17 cells [152]. Most probiotics (such as *L. rhamnosus* GG, *L. casei*, *L. johnsonii* La1, *B. animalis* Bb-12, etc.) are capable of promoting the release of various cytokines and activating macrophages, NK cells, and T cells in a strain-specific and dose-dependent manner, thereby enhancing the intestinal mucosal immune system [153]. SCFAs have also been shown to promote the differentiation of T regulatory cells or effector Th1 cells under different conditions [154]. Additionally, SCFAs upregulate AhR signaling by mediating the production of IL-10 and IL-22 to mitigate intestinal inflammation [155]. Bacterial N-formylated peptides and the pore-forming toxin alpha-hemolysin have been demonstrated to directly activate sensory neurons, thereby regulating inflammation and immune infiltration [156]. Furthermore, bacterial components such as LPS and PGN act as immune agonists and entered the brain through systemic circulation, leading to chronic neuroinflammation and visceral hypersensitivity [157].

The gut microbiota plays a crucial role in shaping the development of the neuroimmune system, and dysbiosis within the microbiota may initiate low levels of inflammation in IBS patients through the neuro-immune network. Consequently, employing drugs designed to prevent and treat certain neurological diseases and neuro-immune disorders may emerge as a potential therapeutic strategy for IBS.

### 4.4. Gut Microbial Metabolites Transferred Directly to the Brain

Certain microbial metabolites originating in the gut can traverse into systemic circulation at varying levels and rates. Recent research indicated that bacterial metabolites derived from polyphenol precursors were present in circulation at levels sufficient to exert biological effects. Furthermore, polyphenols have shown potential in preventing and ameliorating symptoms of IBS by impeding inflammatory signaling pathways [158]. Artichoke polyphenols, for instance, demonstrated the capability to enhance intestinal flora in individuals with IBS and exhibited notable antispasmodic properties [159]. Catechin polyphenols have been found to inhibit the growth of *C. jejuni*, *E. coli*, and *Salmonella* spp., showcasing anti-inflammatory properties and improving symptoms of IBS-related diarrhea [160,161]. In vitro studies have illustrated that polyphenol metabolites could permeate a blood–brain barrier model system, safeguarding neurons through the attenuation of inflammatory responses [162]. As shown in Figure 4, dysbiosis within the microbiome triggers IBS via a neuroimmune network.

The extensive polygenic overlap of IBS with psychiatric and gastrointestinal phenotypes was found beyond what was revealed by genetic associations, and the genetic loci associated with IBS, including but not limited to a wide range of biological pathways in the gut–brain axis, may underlie clinical subtypes of IBS and may form the basis for the development of individualized interventions. This body of evidence enhances our comprehension of the intricate interactions between the brain and the gut in IBS and paves the way for potential pharmaceutical advancements. While several signaling mechanisms associated with brain–gut–microbiome interactions have been postulated, comprehensive investigations into the distinct contributions of elements like neurotransmitters, bacterial metabolites, and immunoreactive compounds are crucial, especially given the escalating incidence of IBS and the mounting global burden of mental health disorders. Such endeavors will aid in exploring ways to adapt current therapies to cater to the needs of individuals grappling with these conditions.

**Figure 4 biomolecules-14-01643-f004:**
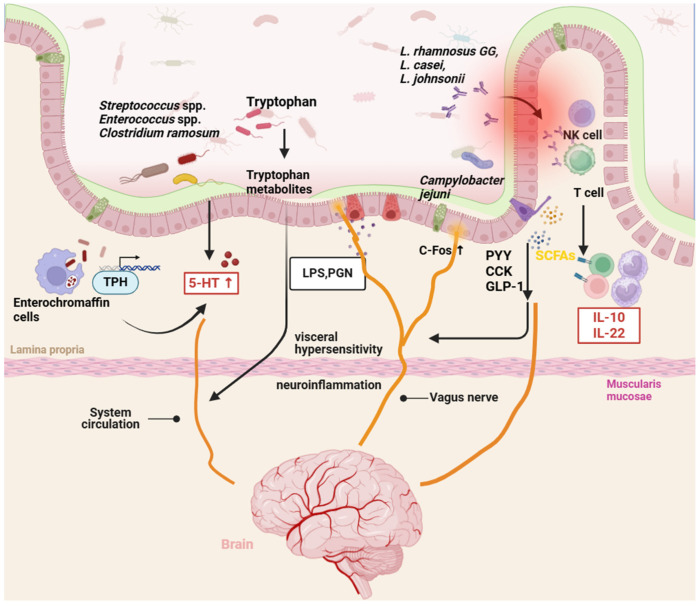
Dysbiosis within the microbiome triggers IBS via neuroimmune networks. Gut microbes directly produce 5-HT or transmit signals to ECs, inducing TPH transcription, promoting 5-HT synthesis, and participating in brain–gut interactions. Intestinal endocrine cells secrete GLP-1, CCK, and PYY in response to stimulation by microbial metabolites, serving as a signal transducer in the brain–gut–microbiome axis. LPS and PGN enter the brain through systemic circulation, leading to chronic neuroinflammation and visceral hypersensitivity. *Campylobacter jejuni* activate the vagus nerve, resulting in increased expression of c-Fos. *L. rhamnosus GG*, *L. casei*, *L. johnsonii*, etc., activating macrophages, NK cells, and T cells, thereby enhancing the intestinal mucosal immune system.

## 5. Gut Microbiome-Mediated Changes in Intestinal Barrier Function and IBS

The gastrointestinal tract of mammals harbors a rich array of immune cells and symbiotic microorganisms, necessitating barrier mechanisms to maintain a delicate balance between them. Within this intricate system, the gut comprises multiple layers of defense [163,164]. Firstly, intestinal secretions and products from symbiotic bacteria are not conducive to orchestrate the colonization of pathogens. Secondly, the mucin polymer barrier, produced by goblet cells, includes antimicrobial peptides (AMPs) from Paneth cells and secretory IgA (sIgA) from intestinal endocrine cells, effectively preventing direct contact between pathogens and epithelial cells. Thirdly, the defense line is reinforced by intestinal epithelial cells, which safeguard the body by impeding harmful substances through tight junctions, adherence junctions, and desmosome junctions, while also facilitating nutrient absorption and metabolism [165]. Furthermore, immune cells (such as dendritic cells, mast cells, macrophages, etc.) located in mucosal membranes and the lamina propria contribute to protection by secreting immunoglobulins, AMPs, and cytokines [166,167]. Published studies have documented increased intestinal or colon permeability in patients with IBS and a positive association between symptoms such as abdominal pain and changes in bowel function [168]. In the context of IBS, intestinal symbiotic bacteria play a pivotal role in upholding intestinal barrier function. Initially, these bacteria thwart the colonization of pathogenic microorganisms by competing for binding sites [169] and stimulate intestinal cells to secrete mucins and AMPs [170]. Additionally, symbiotic microorganisms aid in preserving host intestinal barrier function by either directly or indirectly secreting bioactive molecules [171,172].

### 5.1. Mucus Barrier

This mucous layer serves as a barrier that prevents direct contact between bacteria and epithelial cells, while also serving as the natural habitat for symbiotic bacteria [173,174]. Goblet cells in the gut secrete MUC2, a crucial component essential for disease prevention and occurrence [174,175].

Patients with IBS-D commonly exhibit a damaged intestinal epithelial barrier, similar to those with IBD. This damage resulted in the accelerating degradation of intestinal mucus, leading to increased infiltration of bacteria and toxic components into the intestinal cavity, heightened recruitment of inflammatory cells, and abnormal goblet cell secretion [176]. Colon biopsies from IBS patients have often revealed reduced mucin content in goblet cells [177].

Microbial metabolites, such as meprin β, play a role in promoting the dissociation of MUC2 from goblet cells, releasing it into the intestinal lumen to maintain intestinal homeostasis [178,179]. Various muco-degrading bacteria, including *Akkermansia muciniphila*, *Ruminococcus* spp., *Clostridium* spp., *B. bifidum*, and *B. fragilis*, utilized mucinoglycans as nutrients. Some of these bacteria, like *B. dentium*, could influence the synthesis and elimination of intestinal mucus [180].

Certain mucus-degrading bacteria could alter the protective function of the mucous layer by affecting its pH, viscosity, and elasticity, thereby impacting the colonization of intestinal epithelial cells [181]. Toxins produced by mucin-degrading microorganisms, such as the *B. fragilis* group, *Clostridia*, and *Enterobacteria*, could dissolve mucosal glycoproteins, leading to changes in the intestinal microenvironment and exacerbate intestinal inflammation, abdominal pain, and diarrhea [182]. *A. muciniphila* reduced colonic hypersensitivity and anxiety-like behavior and memory defects in IBS mice, accompanied by a reinforcement of intestinal barrier function [169].

By increasing mucus production and competitively binding to mucin sites, myogenic bacteria prevent pathogen invasion [170]. However, further investigation is necessary to elucidate the underlying mechanisms. Overall, these findings highlight the intricate relationship between intestinal microbes and the regulation of intestinal mucus synthesis and characteristics. The catabolism of host mucinoglycan mediated by bacteria has implications for microbial ecology and intestinal health. The degradation of mucoglycans occurs through a cross-feeding symbiotic network, and the actual diversity of enzymes used to degrade mucoglycans may be greater than currently known. More studies are needed in the future to identify glycosidases targeting mucoglycans, to further understand the microbial degradation mechanisms of host mucoglycans, and to develop new treatments to prevent inflammation and infectious diseases caused by intestinal pathogens.

### 5.2. AMPs and lgA

AMPs are natural antibiotics that exhibit antibacterial properties and regulate the immune system. They are primarily produced by intestinal epithelial cells, Paneth cells, and immune cells [183]. These peptides include human defensins (α-defensins and β-defensins), Cathelicidin, regenerating proteins, lysozymes, and lactoferrin, among others. AMPs play a crucial role in regulating the interactions between the host and gut microbiome in dynamic bidirectional processes. Defensins, particularly abundant in the gut, form pores in bacterial membranes, leading to bacterial death and serving as a vital defense barrier against pathogens. In cases of intestinal inflammation, the expression of human β-defensin 2 significantly increases [184]. Studies have indicated elevated levels of β-defensin 2 in patients with IBS, suggesting the activation of the innate immune system without macroscopic inflammation [185,186].

Cathelicidins possess antibacterial and immunomodulatory properties that enhance epithelial barrier integrity [187]. These peptides, encoded by the gene Cnlp, are essential components of the innate antimicrobial defense in the colon [188]. Regenerating proteins are a group of soluble lectins, which interact with bacterial components, strengthening the intestinal mucosa against bacterial invasion [189]. The dysregulation of AMP production is associated with intestinal microecological imbalances, impacting the secretion of key peptides like α-defensins and RegIII-γ. Dominant-negative mutant of MyD88 mice showed increased symbiotic bacterial translocation and decreased α-defensins and RegIII-γ secreted by Paneth cells [190].

Certain probiotic strains could also produce AMPs that target bacterial membranes, disrupting essential cellular processes. For example, *E. faecium* strains could produce bacteriocins with demonstrated therapeutic potential in inhibiting pathogenic bacteria growth [191]. These bacteriocins, along with other AMPs, have shown promise in regulating intestinal flora and reducing inflammation in conditions like IBS [192].

Immunoglobulin A (IgA) is a crucial antibody present in the intestinal mucosa, existing in dimeric and secretory forms. IgA plays a significant role in trapping bacteria, limiting the colonization of pathogens [193]. High-affinity IgA promoted bacterial clumping, aiding in pathogen clearance from the intestine [194]. Studies suggest that IgA could target both harmful and beneficial bacteria, influencing the development and treatment of diseases, including IBS [195].

sIgA is the primary form of IgA in the intestinal lumen during homeostasis, which is crucial for mucosal immunity and intestinal balance [180]. sIgA recognized specific bacterial surface components, preventing tissue damage and inflammation [196,197]. Disruption in sIgA levels has been linked to conditions like intestinal inflammation, emphasizing its role in maintaining a healthy gut microbiome [198]. The interplay between the intestinal microbiome and IgA production highlights the importance of studying their impact on conditions like IBS [199]. The intricate interactions between the gut microbiota and antibacterial proteins constitute a dynamic dialogue that shapes the delicate balance within the gut ecosystem. The challenges posed by microbial resistance to AMP-induced killing highlight the need for innovative strategies to enhance the effectiveness of antimicrobial defense mechanisms. Future research should provide greater insight into the complex regulatory networks that control AMP expression, microbial resistance mechanisms, and the effects of dysregulation, providing valuable insights into the complex balance required for intestinal homeostasis.

### 5.3. Mechanical Barrier of the Intestinal Epithelium

Below the mucous layer lies an interfacial boundary composed of a single layer of intestinal epithelial cells. These cells are interconnected in successive layers by adhesion connexins, which encompass tight junction (TJ) proteins, adhesion junction (AJ) proteins, gap connexins, and desmosomes [183]. The disruption, whether functional or physical, of TJs, AJs, gap connexins, and desmosomes could result in increased intestinal permeability, leading to dysregulated translocation/transport of inflammatory mediators, potentially culminating in chronic intestinal inflammation. TJ proteins, such as ZO, Occludin, Claudins, Tricellulin, and JAM, create continuous barriers between intestinal epithelial cells, facilitating interaction between luminal components and the epithelium. These proteins serve as a robust physical defense mechanism for submucosal passages [166,167].As shown in Figure 5, gut microbes help protect host intestinal barrier function by directly or indirectly secreting bioactive molecules. 

In IBS-D patients, the loss of the TJ complex allows unrestricted development of the para-cellular pathway, enabling macromolecules and even microorganisms to breach the mucosal layer and come into contact with submucosal neurons, triggering visceral hypersensitivity [200]. Furthermore, IBS-M patients exhibit elevated colonic pro-inflammatory cytokines and decreased Occludin expression [201]. Mice induced with IBS-D through 4% acetic acid displayed intestinal barrier damage and significantly reduced levels of Occludin, Claudin-1, and ZO-1 [202].

In comparison to healthy mice, mice that received an FMT from IBS-D patients demonstrated innate immune activation and compromised intestinal barrier function [203], underscoring the crucial role of the intestinal microbiome in IBS-D pathogenesis. IBS-D mice showed endoplasmic reticulum stress in intestinal epithelial cells and intestinal barrier damage, along with intestinal flora disturbance [204]. Certain symbiotic intestinal bacteria contribute to host health by directly or indirectly bolstering the intestinal barrier [205]. For instance, studies have shown that *L. reuteri* can stimulate intestinal epithelial regeneration and repair a damaged intestinal tract by activating the Wnt/β-catenin pathway [206]. *L. acidophilus* LA1 enhances intestinal tight junction barrier function by upregulating TLR2 membrane expression [207]. Additionally, metabolites from *L. rhamnosus* GG help maintain intestinal barrier homeostasis by reducing FITC- Dextran (FD4) flux and increasing the expression of Occludin and ZO-1 [208]. Metabolites such as SCFAs, indole derivatives, bile acid metabolites, conjugated fatty acids [209], and others produced by intestinal microorganisms have direct effects on the intestinal epithelium, playing a crucial role in regulating intestinal barrier function and enhancing mucosal immunity. Therefore, leveraging beneficial gut microbes or their metabolites appropriately may offer benefits in enhancing intestinal barrier function for patients with IBS [210].

How the pattern recognition of bacterial ligands by the intestinal epithelium translates into impaired intestinal barrier function remains elusive. While colonization with intestinal symbionts is known to cause adaptive changes in intestinal morphology, it has been unclear how microbiome-derived signals are integrated by the immune system into morphogenetic signaling cues in the intestinal epithelial lining. Future studies using primary organoid cultures are needed to address the barrier regulation molecular mechanisms influenced by different symbionts, affecting the structure of villus capillaries and their role in nutrient absorption.

**Figure 5 biomolecules-14-01643-f005:**
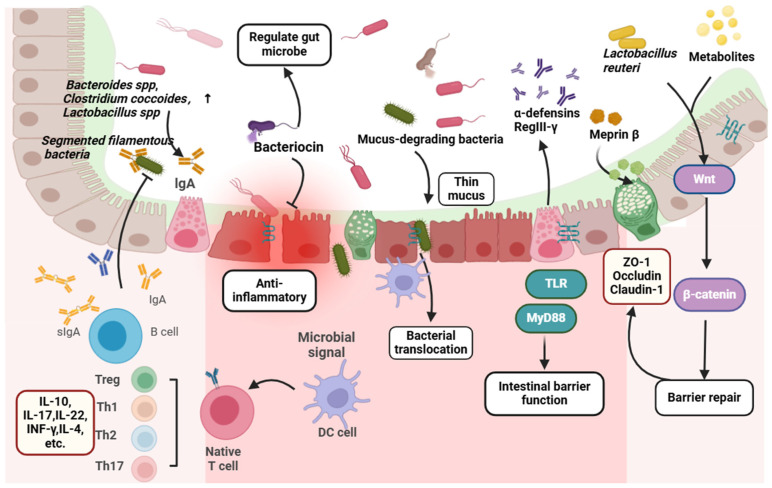
Gut microbes help protect host intestinal barrier function by directly or indirectly secreting bioactive molecules. *Bacteroides* spp., *Clostridium coccoides*, and *Lactobacillus* spp. can promote the secretion of slgA, while slgA can inhibit segmented filamentous bacteria. Gut microbes secrete bacteriocin to inhibit the colonization and growth of pathogenic bacteria and play an anti-inflammatory role. Mucus-degrading bacteria affect barrier protection function and microbial colonization by altering mucus properties. RegIII-γ and α-defensins expression induced by the MyD88 signaling pathway strengthens the intestinal mucosa against bacterial invasion. The microbial metabolite meprin β promotes the dissociation of MUC2 from goblet cells, releasing it into the intestinal lumen to maintain intestinal homeostasis. *L. reuteri* can promote intestinal epithelial regeneration and repair of damaged intestinal tract by activating Wnt/β-catenin pathway. Intestinal macrophages and DCs sense microbial signals, prompting differentiation and activation of T cell subsets (Th1, Th2, Th17 or Treg) to produce cytokines.

## 6. Microbes and Visceral Hypersensitivity

Visceral hypersensitivity primarily refers to heightened pain sensitivity resulting from visceral overreaction to abnormal or even normal external stimuli, often closely associated with the onset of IBS. Compared with healthy individuals, patients with IBS have an increased number of colon mast cells, along with increased degranulation and increased TRPV1 expression. Dysfunctional mast cells, accompanied by the abnormal release of proteases and cytokines (such as histamine, protease, and prostaglandins), could compromise the integrity of the intestinal epithelial barrier, leading to heightened sensitivity of internal organs and disruptions in defecation patterns [211]. Clinical studies have indicated that mast cell stabilizers or histamine receptor antagonists could alleviate symptoms and enhance the quality of life for individuals with IBS [212,213]. The gut microbiota and its metabolites could trigger IBS-related symptoms by activating mast cells and releasing histamine [214]. TRPV1 is a membrane protein and a non-selective cation channel that is mainly expressed in primary afferent nerves and nociceptive sensory neurons. Recently, both clinical and animal studies have shown that TRPV1 is involved in the pathogenesis of IBS visceral hypersensitivity [215]. Recently, clinical and animal studies have shown that TRPV1 is involved in the pathogenesis of IBS visceral hypersensitivity and that changes in the abundance of *Clostridium sensu stricto* 1 and increased sodium butyrate promote visceral hypersensitivity in IBS models through the lincRNA-01028-PKC-TRPV1 pathway [216].

In vitro studies have shown that *Enterococcus faecalis* can inhibit mast cell degranulation in a MyD88-independent manner [217], while the presence of *Salmonella* has been linked to a significant reduction in intestinal barrier proteins and the induction of mast cell degranulation in both IBS patients and healthy subjects [218]. Moreover, rats that received an FMT from IBS patients also exhibited visceral hypersensitivity [219]. Animal and in vitro research has highlighted the potential role of therapies targeting mast cells and the crosstalk between intestinal bacteria in the treatment of intestinal disorders. For instance, studies have demonstrated that mother–child-separated rats displayed visceral hypersensitivity and dysregulation of intestinal fungi, which could be restored to normal levels with specific antibiotic treatments [220]. Miltefosine has shown promise in reducing mast cell degranulation and improving visceral hypersensitivity and microbial dysbiosis in mother–child-separated rats [221].

Clinical studies have shown that microbiota dysbiosis is closely related to functional gastrointestinal diseases and CNS disorders, and depression can amplify visceral sensitivity [222]. Changes in the gut microbiome in patients with mood disorders, an increased ratio of Firmicutes/Bacteroidetes, and probiotic treatment with *Lactobacillus* R0052 and *Bifidobacterium longum* R0175 had beneficial effects on anxiety [223]. The gut microbiota has been implicated in neuropsychiatric disorders by influencing the gene expression pattern of the hypothalamic–pituitary–adrenal (HPA) axis, and the resulting HPA response, such as increased pituitary Crhr1 expression due to a lack of microbiota, is associated with the HPA overresponse observed in GF mice [224]. The CNS also regulates the composition and homeostasis of the gut microbiome (mainly Gram-negative bacteria) through the HPA axis [225]. In addition, the plasticity of glial cells and neurons in the CNS and extensive changes along the brain–gut axis contribute to the development and maintenance of chronic abdominal pain [226]. The ENS is a large network composed of different types of endogenous intestinal neurons and glial cells, including motor neurons, endogenous primary afferent neurons, and interneurons which regulate movement and secretion in a coordinated manner. Studies have shown that active signaling mechanisms between enteric glial cells (EGCs) and neurons regulate gastrointestinal reflexes and that disturbances in their communication may play a role in IBS dysfunction [227]. Studies have shown that the generation of EGC networks parallels the maturation of the gut microbiota (and the immune system) [228] and that the continuous movement of enteric glial cells requires a steady supply of signals from the luminal microbiota. The gut microbiota can regulate the ENS at a fundamental level of structural organization and possible connections, for example, certain microbiome-derived components, may be actively transported through the intestinal epithelium, affecting the EGC. Intestinal neurons express SCFA receptors, and EGC may respond to products produced by the metabolic activity of the microbiome, such as SCFAs. Intestinal epithelial cells may produce mediators that act on EGC in response to the microbiome or its metabolites [229]. Identifying the EGC mediators produced by the intestinal epithelium in the presence of the microbiota and understanding the importance of how the ENS communicates and responds to the gut microbiota and its products are expected to shed light on gut physiology as well as potential therapeutic strategies for IBS.

The limitations of published studies to date, including small sample sizes, poor experimental design, and the absence of biomarkers and specific targeting mechanisms that can be used to determine causal relationships between symptoms, microbiome characteristics, and the enteric and CNS in specific IBS subpopulations, require further study of effective treatment options for this potential mechanism.

## 7. Future Directions

IBS involves interactions between multiple biological systems, and the exploration of its complex pathogenesis is benefiting from the rapid progress of modern science and technology. For example, the rise of intestinal organoid technology, relying on the self-organization of stem cells and the complete replication of functions in vivo, breaks the limitations of animal model research due to species differences and becomes a powerful tool for the current and future study of intestinal functional and inflammatory diseases [230]. Human intestinal organoids are an ideal model system for studying gastrointestinal physiology and immunopathology. Recently, the combination of epithelial organoids and autologous tissue-resident memory T cells with intestinal immune organoids has revealed the mechanism and potential target of intestinal inflammatory response [231]. The multilevel co-culture model of intestinal organoids and bacteria proposed in recent years provides a new perspective for studying microbiota–intestinal epithelium interaction in patients with IBS [232]. The innovative fusion of microfluidic technology and intestinal organoid technology has produced intestinal organ chip technology, which can accelerate the simulation of intestinal epithelial cell absorption, metabolism, and barrier function; establish a bionic model; allow the co-culture of multiple cell types and microorganisms; and integrate mechanical stimulation to provide a more realistic in vitro intestinal microenvironment for studying the efficacy of IBS therapeutic drugs [233]. Intestinal chip technology can expose two of the main barriers (intestinal barrier and blood–brain barrier) to flow and achieve brain–gut crosstalk, which opens up a new way to study the mechanism of the brain–gut axis disorder, IBS [234]. Therefore, organoids and organ-on-a-chip technology can be used to accelerate the development of IBS-related drugs, especially to study the cell-to-cell communication between epithelial cells and other intestinal cell populations, as well as specific interactions with gut microbes, which are critical for maintaining homeostasis and coordinating immune responses. It is hoped that organoid technology can be effectively used to simulate various subtypes and individual differences of IBS, so as to provide diversified models for exploring the mechanism of disease. However, as with any model system, there are limitations and drawbacks to organoids that must be considered, including technical issues, variability/reproducibility issues, and considerations regarding the level of physiological correlation. Organoid transcription profiles may change with culture conditions, growth media, and donor variability, and increased variability can lead to reproducibility problems. At the same time, important experimental details, such as the patient inflammatory status, specific intestinal region, number of organoid cells, and duration of culture, have been omitted from the published literature, which makes the interpretation of data and reproducibility more difficult. This technology is still in its infancy, and we look forward to more researchers focusing on this technology to advance the applicability of organ-on-a-chip systems for studying gastrointestinal physiology and disease.

## 8. Conclusions

In daily life, IBS is a pervasive and heterogeneous disorder, with symptoms like abdominal discomfort, pain, and bloating significantly impacting patients’ quality of life. IBS stands out as the most well-defined and extensively researched functional gastrointestinal disorder. However, developing clinical treatments for IBS poses significant challenges. Firstly, the lack of a universally accepted biomarker complicates drug development efforts. Secondly, the diverse symptoms and progression of IBS contribute to its complexity. Emerging evidence highlights the substantial impact of microbial dysbiosis on the pathogenesis and severity of IBS symptoms and psychological manifestations. Yet, pinpointing the specific biological changes responsible for clinical IBS symptoms remains elusive. Future research endeavors could focus on creating drugs that target particular bacterial enzymes to selectively inhibit the production of harmful metabolites associated with IBS, promote the generation of beneficial metabolites, and minimize unintended side effects. Here, we emphasize the importance of personalized medicine through the regulation of gut microbiota. Based on this, the goal of drug efficacy and access to personalized nutrition can be improved at the individual level. Advances in culturomics and personalized organ-on-a-chip technology, as well as the exponential growth of databases and biobanks of personal information, will make personalized medicine possible. Moreover, personalized dietary interventions favoring the growth of a microbiota capable of producing beneficial chemical signals could offer tailored therapies for individuals with IBS. While the precise dysbiosis characteristic of IBS remains incompletely understood, gut microbes hold promise as prime targets for therapeutic interventions. Unraveling these mechanisms through further research will provide invaluable insights for guiding the development of safe and effective microbial-based treatments for IBS.

**Table 1 biomolecules-14-01643-t001:** Effects of different gut microbiota on IBS.

Classification	Pathway	Effect	Ref.
*Lactobacillaceae*	AhR/IL-22	Restores intestinal permeability and normalizes colon sensitivity, restores cognitive abilities and reduces anxiety-like behaviors	[26]
*Corynebacterium* spp., *Streptococcus* spp., *Enterococcus* spp.	5-HT	Increased intestinal epithelial permeability, visceral hypersensitivity, and immune cell activation	[235]
*E. coli*, *Achromobacter liquefaciens*, *Bacteroides* spp.	Indole	Improve intestinal inflammation and repair mucosal barrier	[48]
*L. reuteri*, *L. bulgaricus* *Alistipes onderdonkii*	Activate the Wnt/beta-catenin and AhR signaling pathway	Regulating intestinal inflammation	[60,61]
*L. reuteri*	Activate histamine H2 receptor	Inhibit intestinal inflammation	[74]
*Fusobacteria*	Production of quercetin, regulate mAChR and 5-HT signaling pathway	Ameliorate chronic constipation induced by loperamide; Improved visceral hypersensitivity in PI-IBS mice	[50,51]
*A. muciniphila*	TLR2 signaling pathway	Improve gastrointestinal motor function	[87]
*B. thetaiotaomicron*	Produced tryptamine to activate G-protein-coupled receptor	Promote the secretion of acetylcholinase and SP and improve colon movement	[62]
*C. butyricum*	TLR2 signaling pathway	Regulate ICC cell viability and promote gastrointestinal motility	[54]
*Lachnospiraceae*	TLR5 signaling pathway	The ability to activate innate immunity in the gut	[56]
*Ruminococcus gnavus*	TAAR1 dependence mechanism	Stimulating gastrointestinal transport	[57]

**Table 2 biomolecules-14-01643-t002:** Effects of gut microbial metabolites on IBS.

Classification		Species	Pathway	Biological and Clinical Correlates	Ref.
SCFAs	Acetate	*Bacteroides* spp.; *Veillonella* spp.	Pyruvate decarboxylated to acetyl-CoA, the Wood–Ljungdahl pathway	Causes symptoms of constipation associated with mucin secretion	[43]
	Propionate	*Fecalibacterium prausnitzii*; *Roseburia* spp.; *Eubacterium rectale*; *Eubacterium hallii*; *Coprococcus comes*	Stimulate PYY and GLP-1 release	Regulate intestinal movement	[85]
	Butyrate		Activation of AMP-activated protein kinase	Enhances the intestinal barrier	[171]
Trp	5-HT	*Corynebacterium* spp.; *Streptococcus* spp.; *Enterococcus* spp.	Stimulate inner sensory neurons; Increase intestinal epithelial permeability; SERT expression decreased	Regulate gastrointestinal motility; Induce IBS symptoms; Visceral pain and degree of hypersensitivity	[48,52,236]
	Kynurenic acid		GPR35	Regulate the gastrointestinal tract and CNS	[237,238]
	Indole		AhR	Improve intestinal inflammation and repair mucosal barrier	[48]
BAs	DCA; LCA	*Bifidobacterium*; *Clostridium*; *E. coli*	Activate TGR	Improve gastrointestinal exercise	[65]
Histamine			Activate H2R	Inhibit intestinal inflammation	[71,74]
Ammonia	Putrescine; cadaverine		Act on chemosensors	Promote intestinal peristalsis	[239]
Proteases	β-glucuronidase	*Alistipes*	Modulating conversion of bilirubin conjugates	Improve intestinal barrier damage and visceral hypersensitivity	[28]
Vitamin	Vitamin D	*Salmonella typhimurium*	NF-κB signaling pathway	Improvement of intestinal barrier	[240,241]
	Vitamin B6	BsS-RS06551		Anti-inflammatory	[242]
Hypoxanthine		*Lachnospiracea strains*; *Barnesiella*; *Prevotella*	Reduced oxygen consumption	Enhance the ability of intestinal cell barrier	[8,240]

## Data Availability

Not applicable.

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
