# Peer review of "The Role of Gut Microbiome in Irritable Bowel Syndrome: Implications for Clinical Therapeutics"

_biomolecules, 2024, doi:10.3390/biom14121643_

Round 1
Reviewer 1 Report
Comments and Suggestions for Authors
The manuscript entitled ”The Role of the Gut Microbiome in Irritable Bowel Syndrome – Implications for Clinical Therapeutics” is, in general, an adequate overview of the current literature related to this topic. It is well written and provides detailed information on various subtopics.
I have some concerns:
IBS still is a heterogenous disorder of which the diagnosis remains to be based on the presence of a set of symptoms. The pathophysiology is multifactorial. What is lacking in the review-manuscript is an interpretation by the authors with respect to the quality of the reported data and the present scientific evidence on the reported data. What is truly evidence based and what are interesting concepts or hypotheses that need to be further explored and proven. I understand that this is a “ review” without the aim of being systematic. However, some classification and judgement is needed. Especially in section 7 on “ Future Directions” such an approach and author’s view is lacking. In my view a scoring of the current scientific evidence of the topics discussed in the review should be undertaken by the authors and will help to improve the quality of the manuscript.
Introduction: excellent introduction to the topic, in general terms, with overview.
Concerning epidemiology: prevalence is depending on the criteria used Rome III or Rome IV. Pooled data from reference 9 point to overall prevalence of IBS based on Rome III criteria of 9.2 %, not 11%.
2.1 microbiota in IBS:
- Add level of evidence of data on probiotics and prebiotics in IBS.
- FMT: Update references to more recent studies such as Holvoet et al , Gastroenterology 2021
- Include data on FMT in IBS from recent reviews and systematic reviews
- potential risks of FMT should be mentioned
- more careful approach of FMT because long term effects of FMT are unknown
2.2 relevant overview presenting associations between IBS subtypes, symptoms , mechanisms and alterations in metabolite concentrations/pathways/pathophysiological mechanisms
2.3 gut microbial composition
This heading is misleading: it is not about the composition ( quantity, diversity) of the gut microbiome but about cellular components of the bacteria
3.1 diet mediated gut microbiome changes in IBS
influences of diet on gut microbiome and on symptoms in IBS : please include recent publications ( Nybacka S. Lancet Gastro Hep 2024 and Van Houte K. et al Gastroenterology 2024) and systematic reviews in your manuscript
Figure 2: junk foods and legumes are presented as identical with respect to high FODMAP content, please specify and explain to the reader what (different) type of FODMAP is present in legumes and in junk food
4. brain gut axis: adequate paragraph (but add level of evidence of findings)
5 gut microbiome, intestinal barrier and IBS: adequate paragraph (but add level of evidence of findings)
6. microbes and visceral hypersensitivity: attention is given only to mast cells and histamine while the role of the enteric and central nervous system are not mentioned. Also include data on TRPV and pain.
7. Future directions: attention is given only to new technology: organoids and organ chip technology. Here a personal (over)view of the authors is lacking. Which mechanisms and pathways are especially relevant ?
8. IBS is not a “common expectation” but a prevalent and heterogenous disorder with complex multi factor pathophysiology. Pay attention to correct English grammar.
Figures: informative and helpful ad obverview of parts of the manuscript text.
Author Response
Thank you very much for your lovely and outstanding comments and suggestions. We have revised our manuscript thoroughly per your opinions. We answer and provide a list of changes to your comments one-by-one as follows.
1.Concerning epidemiology: prevalence is depending on the criteria used Rome III or Rome IV. Pooled data from reference 9 point to overall prevalence of IBS based on Rome III criteria of 9.2 %, not 11%.
Response:
Thank you for pointing this out. We have corrected it in this manuscript.
(P2, L67)
2.1 Microbiota in IBS:
- Add level of evidence of data on probiotics and prebiotics in IBS.
- FMT: Update references to more recent studies such as Holvoet et al , Gastroenterology 2021
- Include data on FMT in IBS from recent reviews and systematic reviews
- potential risks of FMT should be mentioned
- more careful approach of FMT because long term effects of FMT are unknown
Response:
Thank you very much for this comment.
-We have increased the level of data evidence, “A meta-analysis of 82 trials, containing data from more than 10,000 patients had demonstrated showed that potentially effective probiotic compositions (mainly in-cluding Bifidobacterium, Lactobacillus, Enterococcus faecalis, Streptococcu) for improving IBS symptoms had a better improvement effect than single probiotics [34]”.
(P3, L125 and L140-144)
-We have already revised and added reference (Holvoet et al , Gastroenterology 2021) and analyzed. (P4, L155-157)
-We have added data on FMT in IBS from recent reviews and systematic reviews. “A five-year follow-up study at Shanghai No. 10 Hospital showed significant improvements in IBS severity scores, bloating, and abdominal pain after FMT treatment [36]. A meta-analysis of 472 patients with IBS from seven randomized controlled trials showed that FMT significantly relieved patients' discomfort [37].” (P4, L157-161)
-We analyzed several potential risks associated with FMT, long-term efficacy, and future research directions. “There are several potential risks associated with FMT, including the possible transmis-sion of undiagnosed pathogens from the donor, immune responses in some recipients, and unpredictable outcomes due to differences in individual microbiome characteristics. Other issues include immunosuppressive therapy, drug use, and changes in the recip-ient microbiome, which may affect the success and stability of FMT therapy [42]. Strict preoperative screening of donors and adequate preoperative preparation of patients can make FMT a relatively safe treatment. Although FMT has provided significant thera-peutic effects for some patients in the short term, its long-term effects, potential risks, and indications still require further scientific research for confirmation. In clinical practice, FMT should be carefully used under the guidance of doctors according to the specific conditions and treatment needs of patients, while a large number of clinical observations and long-term follow-up are needed to evaluate its role and impact, so as to conduct a comprehensive risk assessment. Further research is needed to identify the beneficial microbiota and the mechanisms involved in ideally transferring a range of well-defined "therapeutic" microbiota and avoiding the risk of introducing potential pathogens. ”(P4, L173-188)
2.2 Relevant overview presenting associations between IBS subtypes, symptoms , mechanisms and alterations in metabolite concentrations/pathways/pathophysiological mechanisms
Response:
Thank you very much for your comments. We increased the metabolite concentration and metabolic changes of different IBS subtypes, and further discussed the differences of metabolites in different IBS subtypes and the contribution of metabolites to IBS. We hope that future studies will focus more on the differences in the metabolic mechanisms of different IBS subtypes and improve IBS symptoms by targeting microbial regulation to obtain beneficial metabolites.
(P4,L195-207; P5 L222-226; P6 L255-259,L264-266,L295-297; P7,309-320)
2.3 Gut microbial composition
This heading is misleading: it is not about the composition ( quantity, diversity) of the gut microbiome but about cellular components of the bacteria
Response:
Thank you for your advice. We corrected it to “Cellular Components of Gut Microbes and the Pathological State of IBS”.
(P7, L321)
3.1 Diet mediated gut microbiome changes in IBS
influences of diet on gut microbiome and on symptoms in IBS : please include recent publications ( Nybacka S. Lancet Gastro Hep 2024 and Van Houte K. et al Gastroenterology 2024) and systematic reviews in your manuscript
Figure 2: junk foods and legumes are presented as identical with respect to high FODMAP content, please specify and explain to the reader what (different) type of FODMAP is present in legumes and in junk food
Response:
Many thanks for the professional suggestion. We have cited ( Nybacka S. Lancet Gastro Hep 2024 and Van Houte K. et al Gastroenterology 2024) and analyzed in your manuscript. (P9, L401-405) At the same time, we added a discussion about the types of FODMAPs that are present in legumes and junk foods. (P10, L420-426)
- Brain gut axis: adequate paragraph (but add level of evidence of findings)
Response:
Thank you for your advice. We add relevant information on the description of IBS and brain-gut axis abnormalities, and discuss the contribution of extensive multigene overlap in psychiatry and gastroenterology to IBS subtypes and personalized interventions, further elucidating the signaling mechanisms associated with brain-gut-microbe interactions. (P11, L482-485; P14, L629-641)
- 5.Gut microbiome, intestinal barrier and IBS: adequate paragraph (but add level of evidence of findings)
Response:
Thank you for your advice. We add to the evidence on the association between IBS and intestinal barrier dysfunction, and discuss future research directions to focus more on microbial degradation of host mucosal sugars, as well as the complex regulatory network of AMP expression, microbial resistance mechanisms and dysregulation, and microbiome derived signaling affecting intestinal epithelium morphogenesis, while exploring more targeted mucosal glycoenzymes. With a view to developing novel treatments that target the gut ecosystem.
(P15, L664-667; P16 L702-708; P17, L747-754; P18 L790-797)
- Microbes and visceral hypersensitivity: attention is given only to mast cells and histamine while the role of the enteric and central nervous system are not mentioned. Also include data on TRPV and pain.
Response:
Thanks for your advice.We add to the evidence that TRPV1 is involved in the pathogenesis of IBS visceral hypersensitivity, and discuss how cross-talk between gut microbiome derived signaling and the enteric nervous system and the central nervous system may influence IBS visceral hypersensitivity.
(P19, L814-816, L823-830, L843-873)
- Future directions: attention is given only to new technology: organoids and organ chip technology. Here a personal (over)view of the authors is lacking. Which mechanisms and pathways are especially relevant ?
Response:
Thanks for your advice. We added the discussion part of the content in this subsection, hoping to further clarify the intercellular communication between epithelial cells of IBS patients and other intestinal cell populations, as well as the specific interaction with intestinal microbes through organo-chip technology, and effectively use organoid technology to simulate various subtypes and individual differences of IBS, so as to provide diversified models for exploring disease mechanisms. In addition, it is also emphasized that the technology is still in its infancy and there are some limitations, and it is expected that researchers will pay attention to it and promote the applicability of organ-on-a-chip system in gastrointestinal physiology and disease research.
(P20, L900-916)
- IBS is not a “common expectation” but a prevalent and heterogenous disorder with complex multi factor pathophysiology. Pay attention to correct English grammar.
Response:
Thank you for pointing this out. We have deleted the incorrect expression in the manuscript and corrected it to “In daily life, IBS is a pervasive and heterogeneous disorder, with…”
(P21, L918)
Figures: informative and helpful ad obverview of parts of the manuscript text.
Response:
Thank you very much for your careful examine and approve manuscripts.
Reviewer 2 Report
Comments and Suggestions for Authors
I congratulate the Authors for their work
The paper is a huge work, but enjoyable to read
I have some suggestion to enhance the quality of the paper
line 66 the word thereby must be written without capital letter
line 70 and following: for a better comparison, I suggest to the Authors to use a single currency
line 135 many strains of L. acidophilus do exist. It would be better indicate which ones were taken into account
line 151 please use capital letter for Methanobrevibacter
line 265 You write "Elevated intestinal proteolytic activity", but is not clear if you mean due to pancreatic enzymes or linked to bacterial proteins. Please specify.
line 327 nice figure
line 361 please specify that is recommended to adopt low fodmap diet for a short period, followed by a partial reintroduction of food high in fodmap. The exclusion of these foods for long periods can alter microbiota, including a reduction in Akkermansia.
line 388 You write about a potential negative role of iron. I suggest you add a sentence about effect of meat and fish, foods rich in this metal
line 465 again, please specify which strain of bacteria
line 474 maybe you forgot a letter in synthesis
line 490 LP 128 does reduce anxiety in mice, according to the study. Please correct
line 498 Please insert the refs for this statement. I suggest these https://pmc.ncbi.nlm.nih.gov/articles/PMC7659911/ and https://pubmed.ncbi.nlm.nih.gov/19630576/
line 534 please specify that probiotics promote the release of cytokines
line 535 I suggest using commas instead of semicolons
line 641 replace "two" with "2"
line 706 nice figure
Author Response
Thank you very much for your lovely and outstanding comments and suggestions. We have revised our manuscript thoroughly per your opinions. We answer and provide a list of changes to your comments one-by-one as follows.
- line 66 the word thereby must be written without capital letter
Response:
Thanks for your advice. We have reviewed the manuscript and corrected the error description.
(P2, L66)
line 70 and following: for a better comparison, I suggest to the Authors to use a single currency
Response:
Thank you very much for your comment. We revised this sentence as “ The direct medical expenses associated with IBS are substantial. For instance, in the UK, annual direct medical costed for IBS can reach as high as ¥1.14 billion to ¥1.76 billion (based on an exchange rate of 1 GBP = 8.74 CNY) [10].In European countries, the total cost (direct and indirect) linked to IBS amounts to up to ¥62.4 billion (based on an exchange rate of 1 EUR = 7.8 CNY) [11] ,Similarly, in China, the management of IBS incurred a total cost of approximately ¥12.383 billion.”
(P2, L71-74)
3.line 135 many strains of L. acidophilus do exist. It would be better indicate which ones were taken into account
Response:
Thanks for your advice. We have given a detailed overview of the strains and functions of Lactobacillus acidophilus, “such as Lactobacillus acidophilus NCFM, can modify the expression of pain-associated receptors, such as μ-opioid and cannabinoid receptors, in the GI tract in mice and hu-mans, thereby improving the symptoms of abdominal pain [33]”
(P3, L138-140)
- line 151 please use capital letter for Methanobrevibacter
Response:
Thanks for your advice. We have modified it.
(P4, L165)
5.line 265 You write "Elevated intestinal proteolytic activity", but is not clear if you mean due to pancreatic enzymes or linked to bacterial proteins. Please specify.
Response:
Thanks for your advice. We revised this sentence as “Elevated intestinal proteolytic activity driven by host serine proteases can disrupt the intestinal barrier integrity,…”
(P6, L301-302)
6.line 327 nice figure
Response:
Thank you very much for your careful examine and approve manuscripts.
7.line 361 please specify that is recommended to adopt low fodmap diet for a short period, followed by a partial reintroduction of food high in fodmap. The exclusion of these foods for long periods can alter microbiota, including a reduction in Akkermansia.
Response:
Thank you for your advice. We agreed with you very much, so we added “The duration of a low-FODMAP diet is generally 3 to 6 weeks, and high-FODMAP foods should be gradually reintroduced after short-term adoption of a low-FODMAP diet. Because long-term low-FODMAP diets may alter the gut microbiome and even reduce Akkermansia [107].”
(P9, L413-416)
8.line 388 You write about a potential negative role of iron. I suggest you add a sentence about effect of meat and fish, foods rich in this metal
Response:
Thank you very much for your advice. We agreed with you very much, so we added “Additionally, iron-rich foods such as meat and fish may influence gut microbiota composition, potentially exacerbating this imbalance. ”
(P10, L450-452)
- line 465 again, please specify which strain of bacteria
Response:
Thank you very much for your advice. We corrected similar mistakes in the whole manuscript.
(P12, L531)
10.line 474 maybe you forgot a letter in synthesis
Response:
Thanks for your advice. We added the letter as follows.
(P12, L540)
11.line 490 LP 128 does reduce anxiety in mice, according to the study. Please correct
Response:
Thanks for your advice. We corrected the error description to “Studies have indicated that L. plantarum PS128 reduced anxiety-like behavior by increasing dopamine and 5-HT concentrations in the brain's striatum in mice”.
(P13, L556)
12.line 498 Please insert the refs for this statement. I suggest these
Response:
Thanks for your advice. We added reference as follows. (BektaÅŸ A,Turk J Gastroenterol. 2020; Berger M,Annu Rev Med. 2009)
(P13, L561)
- line 534 please specify that probiotics promote the release of cytokines
Response:
Thanks for your advice. we revised this sentence as “Most probiotics (such as L. rhamnosus GG, L. casei, L. johnsonii La1, B. animalis Bb-12, etc.) are capable of promoting the release of various cytokines and activating macrophages, NK cells, and T cells in a strain-specific and dose-dependent manner, thereby enhancing the intestinal mucosal immune system”.
(P14, L600-601)
- line 535 I suggest using commas instead of semicolons
Response:
Thanks for your advice. We corrected similar mistakes in the whole manuscript.
(P13, L595-610)
15.line 641 replace "two" with "2"
Response:
Thanks for your advice. We changed the content to “defensin 2 ”
(P17, L718)
16.line 706 nice figure
Response:
Thank you for your careful review.
Round 2
Reviewer 1 Report
Comments and Suggestions for Authors
All the questions and items I have mentioned in my comment have been answered adequately and the text in the manuscript has been changed in line with the answers to my questions /items.